# Determining the Optimal Stimulation Sessions for TMS-Induced Recovery of Upper Extremity Motor Function Post Stroke: A Randomized Controlled Trial

**DOI:** 10.3390/brainsci13121662

**Published:** 2023-11-30

**Authors:** Yichen Lv, Jack Jiaqi Zhang, Kui Wang, Leilei Ju, Hongying Zhang, Yuehan Zhao, Yao Pan, Jianwei Gong, Xin Wang, Kenneth N. K. Fong

**Affiliations:** 1School of Rehabilitation Medicine, Binzhou Medical University, Yantai 264000, China; 2Department of Rehabilitation Medicine, Clinical Medical College, Yangzhou University, Yangzhou 225001, China; 3Department of Rehabilitation Sciences, The Hong Kong Polytechnic University, Kowloon, Hong Kong SAR, China; 4Department of Medical Imaging, Clinical Medical College, Yangzhou University, Yangzhou 225001, China

**Keywords:** ischemic stroke, transcranial magnetic stimulation, motor cortex, upper extremity, magnetic resonance imaging

## Abstract

To find out the optimal treatment sessions of repetitive transcranial magnetic stimulation (TMS) over the primary motor cortex (M1) for upper extremity dysfunction after stroke during the 6-week treatment and to explore its mechanism using motor-evoked potentials (MEPs) and resting-state functional magnetic resonance imaging (rs-fMRI), 72 participants with upper extremity motor dysfunction after ischemic stroke were randomly divided into the control group, 10-session, 20-session, and 30-session rTMS groups. Low-frequency (1 Hz) rTMS over the contralesional M1 was applied in all rTMS groups. The motor function of the upper extremity was assessed before and after treatment. In addition, MEPs and rs-fMRI data were analyzed to detect its effect on brain reorganization. After 6 weeks of treatment, there were significant differences in the Fugl-Meyer Assessment of the upper extremity and the Wolf Motor Function Test scores between the 10-session group and the 30-session group and between the 20- and 30-session groups and the control group, while there was no significant difference between the 20-session group and the 30-session group. Meanwhile, no significant difference was found between the 10-session group and the control group. The 20-session group of rTMS decreased the excitability of the contralesional corticospinal tract represented by the amplitudes of MEPs and enhanced the functional connectivity of the ipsilesional M1 or premotor cortex with the the precentral gyrus, postcentral gyrus, and cingulate gyrus, etc. In conclusion, the 20-session of rTMS protocol is the optimal treatment sessions of TMS for upper extremity dysfunction after stroke during the 6-week treatment. The potential mechanism is related to its influence on the excitability of the corticospinal tract and the remodeling of corticomotor functional networks.

## 1. Introduction

Ischemic stroke is one of the major causes of upper extremity motor dysfunction and is characterized by the sudden loss of focal blood circulation in the brain, with high rates of incidence and a high recurrence rate [1]. Around 2 million people have a stroke in China every year, and around 85% of these patients present with upper extremity motor dysfunction and impairments in independent activities of daily living [2]. Due to the delicate and complex characters of the upper extremity, conventional rehabilitation therapies, including neurodevelopmental therapy [3], motor relearning [4], and task-oriented training [5], have limited effects on improving upper extremity motor function through the indirect promotion of brain functional restructuring. Therefore, developing novel therapeutic strategies to directly promote brain functional restructuring and effectively improve upper extremity motor function after stroke remains a focal point in clinical research.

Repetitive transcranial magnetic stimulation (rTMS) is a newly developed method that directly induces neural plasticity non-invasively. There has been extensive clinical evidence supporting the value of rTMS in improving neurological deficits after stroke [6], and rTMS has been widely used in the rehabilitation of upper extremity motor function [7,8]. Low-frequency rTMS (typically 1 Hz) [9,10,11] could inhibit hyperexcitability in the contralesional cerebral hemispherical cortex, while high-frequency rTMS (typically ≥5 Hz) [12] could reverse hypoexcitability in the ipsilesional cerebral hemisphere after stroke. This inter-hemispheric imbalance between the bilateral primary motor cortex (M1) post stroke contributes to upper extremity motor dysfunction [13]. Many studies found that low-frequency rTMS over the contralesional M1 is an effective therapeutic method for improving motor outcomes in the hemiplegic upper extremity after stroke by modulating the inter-hemispheric imbalance [6,7,8].

Though numerous rTMS-based clinical trials have been reported, there is still no consensus on its optimal therapeutic protocol. For example, rTMS is effective in improving upper extremity dysfunction, but what conditions the maximum therapeutic effect can achieve still need to be further clarified in the treatment protocol, including the number of rTMS pulses, treatment frequency, treatment sessions, etc. Among these, treatments sessions are a vital influencing factor, and clinicians lack objective evidence to choose the suitable treatment sessions of rTMS. At the same time, it has been reported that the therapeutic effect of rTMS is gradually weakened with the extension of time [14]. Therefore, the clinical applications of rTMS in post-stroke motor recovery are limited, and it is an urgent need to determine the optimal treatment sessions in ischemic stroke for the rehabilitation of upper limb function. In addition, the exact neural mechanisms underlying the effects of rTMS in the recovery of upper extremity motor function post stroke are not fully well understood.

Therefore, we hypothesized that rTMS treatments with the same stimulation site, stimulation frequency, and different treatment sessions may have differential effects on upper extremity dysfunction. It is not the case that the longer the treatment sessions, the more effective they are. This study aimed to find out the optimal treatment sessions of TMS for upper extremity dysfunction after stroke during the 6-week treatment and to explore the mechanism of the optimal treatment sessions using motor-evoked potentials (MEPs) and resting-state functional magnetic resonance imaging (rs-fMRI). The results of this study will help doctors and therapists to timely evaluate treatment effect and provide more efficient treatment protocols for patients with stroke.

## 2. Materials and Methods

### 2.1. Study Design

The study had a prospective, single-center, randomized, and parallel-controlled design. All participants received 6 weeks of conventional rehabilitation treatment. Participants were assigned to the control group, the 10-session rTMS group, the 20-session rTMS group, or the 30-session rTMS group. A standard rTMS protocol (1200 pulses, 1 Hz, hotspot of the abductor pollicis brevis muscle of the contralesional hemisphere, and an intensity of 80% of the resting motor threshold) was applied to patients allocated to the rTMS groups one session per day, 5 sessions per week, lasting for 2 weeks, 4 weeks, or 6 weeks, respectively. To ensure effective improvement in brain plasticity using rTMS, conventional physical therapy was performed immediately within 30 min after the end of rTMS stimulation [15]. The treatment plan was based on the assessment at admission, which was aimed at improving muscle tone, muscle strength, balance, coordination, and activities of daily living. Each participant signed an informed consent form before treatment. Participants were informed of the purpose of the research, the procedures involved, the expected completion time, potential adverse reactions, and potential benefits. The study was registered with the Chinese Clinical Trial Registry (ChiCTR2100045249) on 9 April 2021.

### 2.2. Participants

Stroke patients hospitalized in the Department of Rehabilitation of Clinical Medical College in Yangzhou University (Northern Jiangsu People’s Hospital) from June 2021 to June 2022 were enrolled as participants. The inclusion criteria were as follows: (1) ischemic stroke confirmed by neuroimaging (performed by two doctors in the department of rehabilitation medicine, based on the neuroimaging information, including lesions with a high signal on the diffusion-weighted MRI); (2) no history of prior stroke and a disease course of 2–4 weeks; (3) cranial computed tomography and MRI examination; (4) age 40–75 years with no restrictions on sex; (5) Montreal Cognitive Assessment score ≥ 27 points and the ability to cooperate with rehabilitation training and express training feelings clearly; (6) upper extremity motor function: Brunnstrom stage I–III, with muscle tone (modified Ashworth scale) less than 1 (not more than 1+ at the end of treatment); (7) confirming through MEP that the contralesional hemisphere side of the stroke was disinhibited to ensure that participants applied low-frequency rTMS [16]; (8) unilateral onset and a volume of ischemic necrosis between 20 and 40 mL as determined using diffusion-weighted imaging [17]; and (9) voluntary participation in this study and provided informed consent. The exclusion criteria were as follows: (1) severe disorders of consciousness, aphasia, cognitive or communication disorders, or inability to cooperate; (2) comorbid severe cardiopulmonary liver or kidney dysfunction; (3) history of epilepsy, idiopathic epilepsy in first-degree relatives, or use of epileptogenic drugs (such as β-lactams, fluoroquinolones, and macrolides, etc.) [18]; (4) pacemakers, intracranial metal implants, or skull defects; and (5) any other factors that affected assessment or treatment.

### 2.3. Sample Size Calculation

The main purpose of the present study was to determine the optimal treatment sessions of TMS for upper extremity dysfunction after stroke during the 6-week treatment. The calculation of the sample size was based on data from previous research [19]. As the study involved four groups, the calculating formula of sample size was as follows:(1)$$n=2\left(\sigma \frac{z_{1-\alpha /(2\tau )}+z_{1-\beta }}{\mu_{A}-\mu_{B}}\right)$$
(2)$$1-\beta =\Phi \left(z-z_{1-\alpha /(2\tau )}\right)+\Phi \left(-z-z_{1-\alpha /(2\tau )}\right)$$
(3)$$z=\frac{\mu_{A}-\mu_{B}}{\sigma \sqrt{\frac{2}{n}}}$$

In the above formula, n is the sample size, σ is the standard deviation, Φ is the standard normal distribution function, α is Type I error, τ is the number of comparisons to be made, and β is Type II error, meaning 1 − β is the power. We based our sample size calculation on the Fugl-Meyer Assessment for upper extremity (FMA-UE) results of previous pre-experiments, where $\mu_{A}$ = 11 (average of group A), $\mu_{B}$ = 18 (average of group B), σ = 6.0, τ = 3, α = 0.05, and β = 0.20. The calculated required sample size was 16 participants for each group. Assuming a dropout rate of 10%, 18 participants per group (total of 72 participants) needed to be recruited. At last, 72 participants were randomly assigned by a computer-based random number generator to the control, the 10-session rTMS, the 20-session rTMS, or the 30-session rTMS groups in a 1:1:1:1 ratio.

### 2.4. Motor Function Assessment

Motor function assessments of all participants were performed using the FMA-UE [14], the Wolf Motor Function Test (WMFT) [20], the modified Ashworth scale, and the Brunnstrom staging [21]. The FMA-UE, the Wolf Motor Function Test, and the Brunnstrom staging were used to assess upper extremity motor function, including strength, motor control, and coordination, etc. The modified Ashworth scale was used to assess the muscle spasticity of the affected upper extremity.

### 2.5. Repetitive Transcranial Magnetic Stimulation and MEP Measurement 

rTMS was performed using a transcranial magnetic stimulator (CCY-I/IN; Wuhan Yiruide Medical Equipment New Technology Co., Ltd., Wuhan, China) and a figure-eight coil (maximum magnetic field intensity = 2 T, diameter = 9 cm; YRD; Wuhan Yiruide Medical Equipment New Technology Co., Ltd.). The participants laid on the treatment bed. Surface-recording electrodes with a diameter of 1 cm were placed on the abductor pollicis brevis muscle. The reference electrode was placed at the tendon of the abductor pollicis brevis, and the ground wire was connected to the wrist. The stimulation target was the contralesional or ipsilesional M1 area determined to use the 10–20 EEG system. The optimal scalp position for inducing MEPs in the abductor pollicis brevis muscle was found by moving the coil in steps of 1 cm over the M1 until the largest MEPs were found. The resting motor threshold was defined as the output intensity of the stimulation device when more than 5 out of 10 single-pulse stimulations recorded more than 50 mV on the electrode. To ensure the accuracy of the coil in one session, a specialized therapist confirmed that the coil would not deviate more than 1 cm from the target [22]. A total of 1200 pulses with a frequency of 1 Hz and an intensity of 80% of the resting motor threshold were delivered to the contralesional hemisphere. MEP measurement was performed once every 10 rTMS sessions, and the amplitude of the MEP was recorded to analyze the state of the corticospinal excitability.

### 2.6. Resting-State Functional Magnetic Resonance Imaging Acquisition

To compare changes in functional connectivity mediated by rTMS, each participant was scanned on a 3.0 T MRI scanner (Discovery MR750, GE Medical Systems, Milwaukee, WI, USA) with a standard 8-channel head coil. These participants were scanned for T1-weighted images and rs-fMRI acquisitions within 24 h before and after the intervention. T1-weighted images were used for reconstructing individual brain anatomy and were acquired using the following parameters: pulse repetition time/echo time = 1900 ms/3.39 ms, field of view = 240 × 176 mm, matrix = 256 × 176, slice thickness = 0.9375 mm, flip angle = 7°, reverse time = 1100 ms, scan time = 4 min, and number of layers = 32. T2-weighted rs-fMRI volumes were used for functional connectivity analyses and were acquired using the following parameters: echo-planar imaging sequence = 31 slices, pulse repetition time/echo time = 2000 ms/30 ms, slice thickness = 4 mm, matrix = 64 × 64 mm, field of view = 240 × 240 mm, flip angle = 90°, number of layers = 31, and scanning time = 8 min. Each patient was asked to keep their eyes closed and stay as still as possible during scans. 

### 2.7. MRI Data Preprocessing and Analysis

We used Statistical Parametric Mapping (v12; http://www.fil.ion.ucl.ac.uk/spm; accessed on 1 May 2023) [23] and the RESTPLUS software (V6.1; http://restfmri.net/forum/restplus; accessed on 1 January 2022) [24] to preprocess the rs-fMRI data. First, the first 15 images were removed for each subject to ensure stable data. Then, slice timing was corrected on the remaining data. Next, data were motion-corrected (i.e., aligning each volume to the mean image of all volumes). After evaluating head motion parameters, we ensured that each patient’s head displacement did not exceed 3 mm, and the rotation angle did not exceed 3°. Then, we used Advanced Normalization Tools software28 for spatial normalization. First, we registered each subject’s T1 structural images to the mean image of their functional image and mapped the corresponding lesions to their functional image as well. Second, we registered the T1 structural image to the Montral Neurological Institute and Hospital (MNI) standard space. Then, we applied the nonlinear transformation parameters obtained in the previous step to each motion-corrected volume to obtain each subject’s functional image in the MNI standard space and then resampled the spatially normalized functional image to 3 mm × 3 mm × 3 mm voxels. Finally, all voxels were smoothed using Gaussian smoothing with a full width at half maximum of 6 mm. We performed subsequent denoising steps, including detrending and regressing out noise covariates, which included motion-related parameters (i.e., friston24), white matter signals, cerebrospinal fluid signals, and other confounding variables. The filtering range was 0.01–0.08 Hz.

### 2.8. Statistical Analysis

Statistical analyses were performed using SPSS (v19.0; IBM Corp., Armonk, NY, USA), and the alpha level was set to *p*  <  0.05. Data were assessed for normality using Shapiro-Wilks tests. Patients were compared between groups with one-way ANOVA, with least significant difference post hoc analysis correcting for multiple comparisons [25] of age, time since stroke, education years, FMA-UE scores, and WMFT scores. Means and standard deviations were reported. The Brunnstrom stage and Ashworth scale compared using the Wilcoxon rank-sum test were used, and we reported median and interquartile ranges. Sex, affected side, stroke site, and risk factors were summarized using frequencies and percentages, and groups were compared using the chi-square or Fisher’s exact tests.

The seed-based correlation method was used to measure the functional connectivity of the rs-fMRI [26,27]. The ipsilesional M1 and premotor cortex were selected as the seed regions of the brain, respectively. Two-sided paired *t*-tests were conducted to assess the functional connectivity difference caused by rTMS and conventional rehabilitation treatment (post-intervention minus pre-intervention). The statistical significance threshold was set to 0.01 for uncorrected *p*-values and 0.05 for cluster AlphaSim-corrected *p*-values, and we selected clusters with more than 10 voxels for analysis.

## 3. Results

### 3.1. Clinical and Functional Characteristics

A total of 72 participants were randomly assigned to the 10-session, 20-session, and 30-session rTMS groups and the control group. All participants completed 6 weeks of conventional rehabilitation training, motor function assessment, and MEP measurement before and after treatment. Because of some reasons, some participants did not participate in rs-fMRI examinations (Figure 1).

There were no significant differences in age, sex, injured hemisphere, and other characteristics among all participants (Table 1).

### 3.2. Assessment of Upper Extremity Motor Function

Table 2 and Table 3 show the FMA-UE and WMFT scores of each group at different time points (before and after intervention). At different time points, there were comparisons of the scores between all TMS treatment groups and the control group, as well as the scores between different TMS groups. Table 2 shows that, in inter-group comparisons, there were no significant difference in FMA-UE scores among the groups before treatment (*p* = 0.97). After 2 weeks of treatment, there were significant differences in FMA-UE scores between all TMS treatment groups and the control group (10-session group vs. control group: *p* = 0.005, 20-session group vs. control group: *p* = 0.002, and 30-session group vs. control group: *p* = 0.001), while there were no significant differences between the 10-session group, the 20-session group, and the 30-session group (10-session rTMS group vs. 20-session group: *p* = 0.81; 10-session rTMS group vs. 30-session group: *p* = 0.50, and 20-session rTMS group vs. 30-session group: *p* = 0.66). 

After 4 weeks of treatment, there were significant differences in FMA-UE scores between the 20- and 30-session groups and the control group (20-session group vs. control group: *p* = 0.0001 and 30-session group vs. control group: *p* = 0.00002) and between the 10-session group and the 30-session group (*p* = 0.001), but there were no significant differences between the 10-session group and the control group (*p* = 0.15) and between the 20-session group and the 30-session group (*p* = 0.71). 

After 6 weeks of treatment, there were significant differences in FMA-UE scores between the 10-session group and the 30-session group (*p* = 0.001) and between the 20- and 30-session groups and the control group (20-session group vs. control group: *p* = 0.002 and 30-session group vs. control group: *p* = 0.00004), while there were no significant differences between the 20-session group and the 30-session group (*p* = 0.28) and between the 10-session group and the control group (*p* = 0.45). In intra-group comparisons, aside from the control group having no significant difference between its scores before treatment and after 2 weeks (*p* = 0.02), all other groups showed significant differences compared to before treatment (*p* < 0.001). And the results of WMFT and FMA-UE were similar (Table 3).

### 3.3. Assessment of MEPs

Figure 2 shows that in inter-group comparisons, there were no significant differences in the amplitudes of MEPs that were obtained from the contralesional M1 between the control group and the 20-session group before treatment (*p* = 0.87). However, after 4 weeks of treatment, there were significant differences (*p* = 0.02) between these two groups. In intra-group comparisons, there was no significant difference between the pre- and post-intervention of the control group (*p* = 0.75), while there was a significant decrease in the 20-session group after treatment (*p* = 0.007).

### 3.4. Regions with Increased Functional Connectivity in the 20-Session rTMS Group

We compared differences in imaging data between the 20-session rTMS group and the control group before and after treatment. As shown in Figure 3 and Table 4, with the ipsilesional M1 as the region of interest, compared with the control group, the brain regions with significantly higher functional connectivity in the 20-session rTMS group were the ipsilesional precentral gyrus, ipsilesional postcentral gyrus, ipsilesional cingulate sulcus, contralesional temporal pole and middle temporal gyrus, and contralesional anterior cuneus. With the ipsilesional premotor cortex as the region of interest, compared with the control group, the brain regions with significantly higher functional connectivity in the 20-session rTMS group were the ipsilesional precentral gyrus, ipsilesional rectus gyrus, ipsilesional olfactory cortex, ipsilesional superior occipital gyrus, ipsilesional superior parietal gyrus, ipsilesional cingulate sulcus, contralesional supplementary motor area, and contralesional rectus gyrus.

## 4. Discussion

In the present study, the effects of rTMS on upper extremity motor function in patients with ischemic stroke between different treatment session protocols were observed, and the changes in brain function after treatment were further analyzed. First, rTMS combined with conventional physical therapy could effectively improve upper extremity motor function dysfunction after stroke. After 10 sessions (2 weeks) of treatment, the curative effects of all TMS treatment groups were significantly better than that of the conventional group. Second, 20 sessions (4 weeks) of rTMS applied to the contralesional M1 was an optimal stimulating protocol for the improvement in upper extremity motor function after stroke, compared to 10- and 30-session protocols, during the 6-week treatment. Third, 20 sessions of rTMS decreased the excitability of the contralesional corticospinal tract represented by the amplitudes of MEPs and enhanced the functional connectivity of the ipsilesional M1 or premotor cortex with the the precentral gyrus, postcentral gyrus, and cingulate gyrus, etc. 

Consistent with other studies [13], as shown in the Table 2 and Table 3, rTMS (1 Hz) over the contralesional M1 significantly improved upper extremity motor function after ischemic stroke, which was manifested by the significant increases in FMA-UE and WMFT scores. Before treatment, there were no significant differences among the groups, but after two weeks (10-session), all the groups using rTMS had significant differences in upper extremity motor function with the control group. At the same time, the upper extremity motor function of the groups with a continuous application of rTMS, such as that of the 20-session group at four weeks and that of the 30-session group at four weeks and six weeks, were significantly better than those of the control group in the same period. This all means that within six weeks, the subjects who used rTMS combined with conventional treatment will obtain better treatment effects, compared with those who only used the conventional treatment, after a 10-session treatment.

According to the results of the scale assessments, we found that 20 sessions is the optimal number of treatment sessions within six weeks. The study showed that if only 10 sessions of rTMS treatment are conducted, the superiority of combination therapy in upper extremity motor function will be lost within two weeks after the end of the rTMS treatment. But two weeks after the end of rTMS treatment, the superiority of the combination therapy for upper extremity motor function after 20 sessions of rTMS treatment was still in existence because there were significant differences in upper extremity motor function between the 20-session group and the control group. This means that continuous rTMS combined with conventional treatment would have a better treatment effect than that of the control group, but only 10 sessions of rTMS do not have most of the recovery potential of rTMS during the 6-week treatment. However, continuous treatment does not seem to be the optimal protocol. After six weeks of treatment, there was no significant difference in upper extremity motor function between the 20-session group and the 30-session group. It is worth noting that, compared to before treatment, the FMA-UE and WMFT scores of the 20- and 30-session groups exceeded the minimum clinically significant difference [28], which means that 20-session rTMS combination therapy can make most patients feel obvious changes within six weeks. All showed that the 20-session rTMS combination therapy had the same treatment effect as the 30-session rTMS combination therapy after six weeks. The possible reasons are that with prolonged treatment, the recovery course gradually comes into the biphasic balance recovery stage, and the excitability of the motor cortex on the contralesional hemisphere also decreases, which makes low-frequency rTMS therapy on the contralesional hemisphere less effective for promoting the recovery of brain function [29,30]. In addition, the speed of brain plasticity after stroke is closely related to the disease period, and it has been established in longitudinal studies that rehabilitation in the subacute stage of stroke (1–2 months after onset) is better than in the later stages [31,32]. Combined with the treatment effect and the length of the treatment cycle, it shows that the 20-session group is the optimal treatment session of TMS for upper extremity dysfunction after stroke during the 6-week treatment. 

Based on the above-mentioned information, we compared differences in the amplitude of the MEP and functional connectivity between the 20-session rTMS group and the control group. As shown in the Figure 2, after 4 weeks of treatment, the improvement in the therapeutic effect was accompanied by the decrease in the amplitude of the MEP, which were obtained from the contralesional M1. This is supported by results from previous studies [33,34] showing that low-frequency rTMS on the contralesional M1 could significantly influence the function of the corticospinal tract and result in an improvement in upper extremity motor function after stroke. It was reported that rTMS could modulate the excitability of residual nerve cells in both the premotor cortex and M1 [35]. Further animal studies [36,37,38] have also shown that the potential mechanism behind this is related to the secretion of exciting and inhibiting neurotransmitter releases. However, how this affect network-level brain function is still not quite clear.

The development of MRI provided an efficient way to explore the reorganization of brain function in participants after stroke. Previous studies have shown that normal upper extremity motor function depends on the joint activation and effective connection of bilateral motor brain areas [39,40] or even motor brain networks [41,42,43], while stroke leads to the anatomical and/or functional destruction of the brain, finally resulting in upper extremity motor function dysfunction [44]. In this study, low-frequency rTMS over the contralesional M1 reduced the inhibition of the contralesional hemisphere on the ipsilesional hemisphere, resulting in increased functional connectivity between ipsilesional M1 or premotor cortex and the precentral gyrus and the supplementary motor area, etc., which are associated with some upper extremity motor functions, including muscle strength and tension [13,29]. At the same time, the functional connectivities between the M1 or the premotor cortex and the postcentral gyrus, the superior parietal gyrus, and cingulate gyrus were also enhanced after treatment. Research indicates that the postcentral gyrus is an important area for sensation generation, including touch, pressure, temperature, and pain [45,46], and the superior parietal gyrus and cingulate sulcus are important regions related to memory and executive function that are involved in complex motor functions and motor executions under compensatory conditions [47,48,49]. Therefore, we speculated that the enhanced functional connection between the contralesional M1 or premotor cortex and some specific cerebrum areas in the ipsilesional or contralesional hemisphere reflects the promotion of compensation across cortico-cortical motor function networks after stroke, thus improving the motor function of the upper extremity.

There were several limitations in the present study. First, because of the small sample size, sampling bias was inevitable. Therefore, the results of the study could only be generalized to participants with the same characteristics as the study sample. Second, the lack of follow-up made it impossible to explore the long-term effect of rTMS. Additionally, we only analyzed rs-fMRI data of the control group and the 20-session rTMS group, which limited the comprehensive analysis of dynamic brain functional recombination post rTMS. Further studies should focus on the dynamic and long-term effects of rTMS on brain function after stroke.

## 5. Conclusions

Our study demonstrated that rTMS combined with conventional physical therapy could effectively improve upper extremity motor function dysfunction after stroke. But 20 sessions (4 weeks) of rTMS on the contralesional M1 of combination therapy was the optimal number of stimulation sessions for the improvement in upper extremity motor function after stroke during the 6-week treatment, compared with 10- and 30-session stimulation sessions. The potential mechanism is related to its influence on the excitability of the corticospinal tract and the remodeling of brain functional networks because 20 sessions of rTMS decreased the excitability of the contralesional corticospinal tract represented by the amplitudes of MEPs and enhanced the functional connectivity of the ipsilesional M1 or premotor cortex with the the precentral gyrus, postcentral gyrus, and cingulate gyrus, etc.

## Figures and Tables

**Figure 1 brainsci-13-01662-f001:**
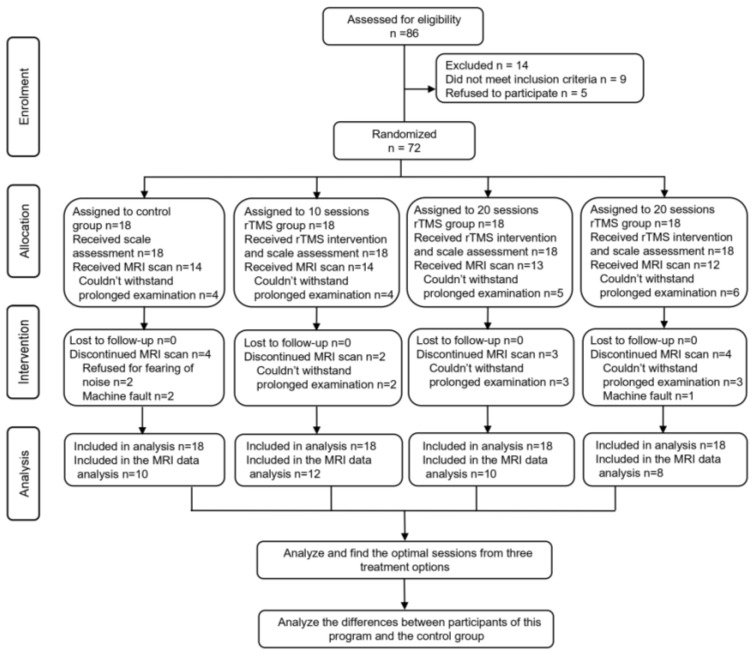
The flow chart of this study.

**Figure 2 brainsci-13-01662-f002:**
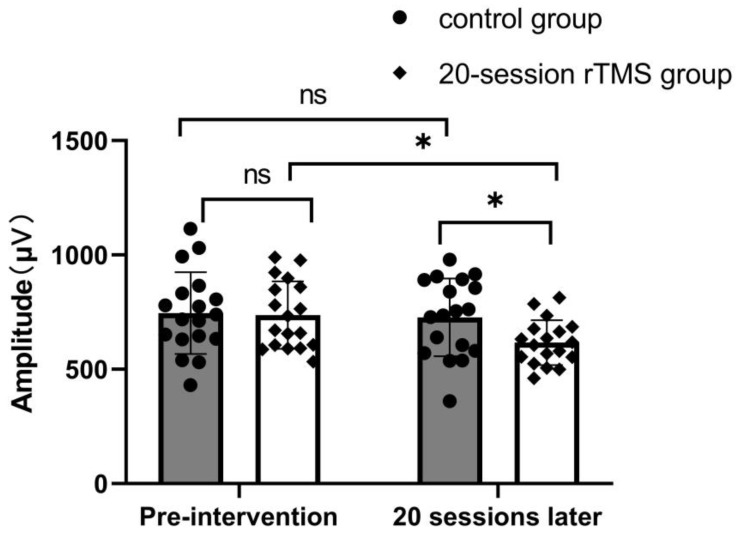
The amplitudes of MEPs in the control group and the 20-session group before and after treatment.* *p* < 0.05; ns *p* > 0.05.

**Figure 3 brainsci-13-01662-f003:**
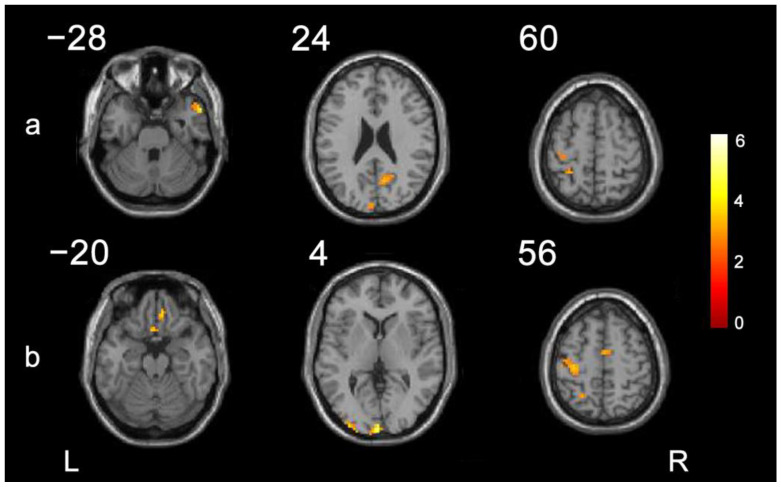
The brain regions with significantly higher functional connectivity increased more in the 20-sessions rTMS group than in the control group. (**a**) Comparison with the affected M1 as the ROI. (**b**) Comparison with PMC as the seed point. L: left; R: right.

**Table 1 brainsci-13-01662-t001:** Characteristics of the participants in the study groups before treatment.

Characteristics	Control Group (*n* = 18)	10-Session rTMS Group (*n* = 18)	20-Session rTMS Group (*n* = 18)	30-Session rTMS Group (*n* = 18)	*p*
Age (years)	57.8 ± 17.6	58.0 ± 12.3	57.3 ± 11.5	57.6 ± 11.7	0.99
Sex (M)	10 (56%)	10 (56%)	11 (61%)	9 (50%)	0.99
Affected hemisphere (L)	10 (56%)	11 (61%)	10 (56%)	10 (56%)	0.98
Stroke site (Supratentorial)	14 (78%)	14 (78%)	15 (83%)	15 (83%)	0.99
Time from stroke onset (d)	18.5 ± 2.3	18.28 ± 2.7	18.22 ± 3.4	18.9 ± 3.6	0.91
Education (years)	12.2 ± 3.6	12.0 ± 3.0	11.3 ± 2.8	11.7 ± 3.2	0.87
History of diabetes	8 (44%)	9 (50%)	7 (39%)	9 (50%)	0.95
History of hypertension	12 (67%)	11 (61%)	11 (61%)	10 (56%)	0.92
Current smoking	3 (17%)	4 (22%)	4 (22%)	3 (17%)	0.99
Current alcohol drinking	5 (28%)	4 (22%)	5 (28%)	4 (22%)	0.99
Brunnstrom stage (upper extremity)	2 (2–3)	2 (2–2)	2 (1–3)	2 (2–2.25)	0.92
Brunnstrom stage (hand)	1 (1–1)	1 (1–2)	1 (1–1.25)	1 (1–2)	0.11
Ashworth scale	0 (0–1)	0 (0–1)	0 (0–1)	0.5 (0–1)	0.98

Data presented as mean (SD), *n* (%), or median (IQR).

**Table 2 brainsci-13-01662-t002:** FMA-UE scores for all participants before and after intervention.

	Control Group (*n* = 18)	10 Sessions TMS Group (*n* = 18)	20 Sessions TMS Group (*n* = 18)	30 Sessions TMS Group (*n* = 18)
Pre-intervention	7.1 ± 3.5	7.4 ± 3.3	7.4 ± 3.6	7.6 ± 2.7
10 sessions later	10. ± 3.9	14.0 ± 3.5 ^a,b^	14.3 ± 4.9 ^a,b^	14.9 ± 4.1 ^a,b^
20 sessions later	13.9 ± 5.1 ^b,c^	16.5 ± 3.5 ^b,c^	21.3 ± 6.0 ^a,b^	22.0 ± 6.4 ^a,b^
30 sessions later	17.5 ± 6.4 ^b,c^	19.1 ± 4.2 ^b,c^	24.2 ± 6.7 ^a,b^	26.4 ± 6.8 ^a,b^

^a^ Compared with the control group of the same treatment phase, *p* < 0.05; ^b^ compared with same group pre-intervention, *p* < 0.01; ^c^ compared with the 30-session rTMS group of the same treatment phase, *p* < 0.05.

**Table 3 brainsci-13-01662-t003:** WMFT scores for all participants before and after intervention.

	Control Group (*n* = 18)	10 Sessions TMS Group (*n* = 18)	20 Sessions TMS Group (*n* = 18)	30 Sessions TMS Group (*n* = 18)
Pre-intervention	6.9 ± 4.5	7.1 ± 2.7	6.6 ± 3.3	7.3 ± 3.8
10 sessions later	10.4 ± 4.8	14.2 ± 5.6 ^a,b^	14.3 ± 4.1 ^a,b^	15.3 ± 6.0 ^a,b^
20 sessions later	14.3 ± 5.8 ^b,c^	16.2 ± 5.7 ^b,c^	19.9 ± 5.2 ^a,b^	22.2 ± 7.1 ^a,b^
30 sessions later	17.4 ± 6.7 ^b,c^	18.3 ± 6.1 ^b,c^	23.9 ± 6.8 ^a,b^	26.9 ± 8.2 ^a,b^

^a^ Compared with the control group of the same treatment phase, *p* < 0.05; ^b^ compared with the same group pre-intervention, *p* < 0.01; ^c^ compared with the 30-session rTMS group of the same treatment phase, *p* < 0.05.

**Table 4 brainsci-13-01662-t004:** Positive functional connectivity after 20 sessions of rTMS.

ROI	Cluster	Connected Region	Peak MNI Coordinates			Peak F Value	Cluster Size
			X	Y	Z		
M1 (ipsilesional side)	1	Precentral_L *	−39	−21	63	2.78	17
	2	Postcentral_L *	−27	−42	60	3.17	12
	3	Postcentral_L *	−15	−39	78	3.88	11
	4	Calcarine_L	0	−99	3	5.52	57
	5	Temporal_Pole_Mid_R	57	12	−27	3.94	13
	6	Precuneus_R	12	−63	24	3.34	34
PMC (ipsilesional side)	1	Precentral_L *	−33	−24	60	3.96	70
	2	Rectus_L,Olfactory_L	0	18	−21	3.62	11
	3	Occipital_Mid_L	−36	−96	0	4.15	23
	4	Parietal_Sup_L	−24	−54	54	3.27	12
	5	Calcarine_L	0	−99	3	5.48	43
	6	Supp_Motor_Area_R	3	−6	60	3.55	16
	7	Rectus_R	9	36	−21	3.55	14

* Clusters in the precentral or postcentral gyrus. L: left; R: right

## Data Availability

Data are contained within the article.

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
