# Peer review of "Determining the Optimal Stimulation Sessions for TMS-Induced Recovery of Upper Extremity Motor Function Post Stroke: A Randomized Controlled Trial"

_brainsci, 2023, doi:10.3390/brainsci13121662_

Round 1

Reviewer 1 Report

Comments and Suggestions for Authors

Lv et al. explored the use of various regimens of rTMS on upper limb motor recovery after stroke. This article is well-written and an interesting topic, but I have several methodological/interpretation questions, which are enumerated below, as well as some general points.

1.      There needs to be a clearly stated hypothesis at the end of the introduction.

2.      There needs to be more detail on the study design. Why were groups separated into 10, 20, and 30 session administrations? Why was it not set up such that everyone got 30 “sessions”, but (1) for the 10-session group, the latter 20 were sham rTMS, (2) for the 20-session group, the latter 10 sessions were sham rTMS, and (3) for the 30-session group, there was no sham. Furthermore, was comparison made between individuals in, say, the 20-session group at the end of their 10th session vs the 10-session group? Theoretically the effect should be the same given that they received the same number of sessions.

3.      Line 98-99, what diagnostic criteria were used for determining stroke via neuroimaging? Who performed the diagnosis?

4.      Line 99: “No history of stroke” – I assume no history of prior stroke? How was this confirmed?

5.      Line 100: Why was either CT or MRI used? They have very different sensitivities to stroke lesions. What field strength was used for the MRI? How many were assessed using each method? Was a subgroup analysis performed on the CT group and the MRI group?

6.      Line 112: How was “epileptogenic” defined, and can you provide some examples of drugs that fell into this category?

7.      Lines 122-124: Several questions here.

a.      What pre-experiments are being referred to? Are they published?

b.      What do the numbers 11, 18, and 6 refer to? What units and context?

c.      Was adjustment for multiple comparisons performed in the sample size calculation (cf. Vickerstaff et al., 2015)?

d.      Why was a one-tailed test used?

8.      Line 130: “Mote” should be “Motor.”

9.      Line 152-163. More details of the MRI paradigm are needed. E.g., what was the field strength?

10.   Line 186-187. Why was no adjustment for multiple statistical tests performed?

11.   Lines 192-193. What does “as appropriate” mean? Why was one not used exclusively?

12.   Tables 2-3. This format is unclear, what is being shown here? Please elaborate more in the text.

Comments on the Quality of English Language

English language quality is adequate

Author Response

  1. There needs to be a clearly stated hypothesis at the end of the introduction.

Response: Thank you for the suggestion. We have added the hypothesis at the end of the introduction. 

Our modifications are as follows:

Therefore, we speculated that rTMS treatments with the same stimulation site and stimulation frequency and different treatment sessions may have different effects on upper extremity dysfunction, and it is not the case that the longer the treatment sessions, the more effective. (Line 79-82)

  1. There needs to be more detail on the study design. Why were groups separated into 10, 20, and 30 session administrations? Why was it not set up such that everyone got 30 “sessions”, but (1) for the 10-session group, the latter 20 were sham rTMS, (2) for the 20-session group, the latter 10 sessions were sham rTMS, and (3) for the 30-session group, there was no sham. Furthermore, was comparison made between individuals in, say, the 20-session group at the end of their 10th session vs the 10-session group? Theoretically the effect should be the same given that they received the same number of sessions.

Response: We thank the reviewer for raising this valuable concern. As the aim of this study was to explore the optimal sessions of rTMS in the treatment upper limb function in ischemic stroke patents, the study groups were set according to different sessions. To exclude the placebo effect which might come from the process of carrying out rTMS, sham stimulation was not applied in the left sessions of the 10- and 20-session groups which was in coincident with clinical operation.  As evidenced by our behavioral outcome in our study, the effect was the same between individuals in the 20-session group at the end of their 10th session and the 10-session group (Table 2 and 3).

  1. Line 98-99, what diagnostic criteria were used for determining stroke via neuroimaging? Who performed the diagnosis?

Response: We thank the reviewer for bringing this to our attention. According to reference, the diagnosis of ischemic stroke was performed by two doctors in rehabilitative department based on the neuroimaging information including lesions with high signal on diffusion-weighted MRI. We have revised it in the text (line 109-111).

  1. Line 99: “No history of stroke” – I assume no history of prior stroke? How was this confirmed?

Response: We thank the reviewer for bringing this to our attention. We have revised it in the text ï¼ˆline 111). We mainly confirmed no history of prior stroke by asking patient the medical history and analyzing no old stroke lesions on neuroimaging.

  1. Line 100: Why was either CT or MRI used? They have very different sensitivities to stroke lesions. What field strength was used for the MRI? How many were assessed using each method? Was a subgroup analysis performed on the CT group and the MRI group?

Response: Thank you for the suggestion. I'm really sorry for the inaccurate expression in the original text. Since most of the stroke patients were first received CT scan to rule out the possibility of cerebral hemorrhage in emergency department in stroke onset followed by the MRI scan for a definitive diagnosis. Correspondingly, all patients in our study were received both CT and MRI scan. We have corrected it in the revised manuscript (line 112).

  1. Line 112: How was “epileptogenic” defined, and can you provide some examples of drugs that fell into this category?

Response: Thank you for the suggestion.In the past, we had found that some drugs can cause brain state changes, making EEG indicate that patients have epileptic tendency, mainly including β-lactams, Fluoroquinolones, and Macrolides (line 124-125). Meanwhile, we have added a new reference:

Wanleenuwat, P.; Suntharampillai, N.; Iwanowski, P. Antibiotic-induced epileptic seizures: mechanisms of action and clinical considerations. Seizure. 2020, 81, 167-174.

  1. Lines 122-124: Several questions here.

a.What pre-experiments are being referred to? Are they published?

Response: Thank you for the suggestion. There are no similar publications and these data are from pre-experiments (several subjects) in the exploration phase of the study.

b.What do the numbers 11, 18, and 6 refer to? What units and context?

Response: We thank the reviewer for bringing this to our attention. We have corrected it in the revised manuscript.11 and 18 are the average FMA-UE scores of the group A and the group B, and 6 is the standard deviation of the two groups. Therefore, these data have no unit.

Our modifications are as follows:

In the above formula, n is sample size, σ is standard deviation, Φ is the standard Normal distribution function, α is Type I error, τ is the number of comparisons to be made and β is Type II error, meaning 1−β is power. Based our sample size calculation on the Fugl-Meyer Assessment for upper extremity (FMA-UE) results of previous pre-experiments that = 11 (average of group A), = 18 (average of group B), σ = 6.0 (Line 138-142).

c.Was adjustment for multiple comparisons performed in the sample size calculation (cf. Vickerstaff et al., 2015)?

Response: We thank the reviewer for bringing this to our attention. Our results were based on the Fugl-Meyer Assessment for upper extremity ( FMA-UE ), so multiple comparison correction was not used and the impact of other indicators was not considered. Thank you for raising this question. We will consider the impact of multiple evaluation factors in subsequent work and correct for multiple comparisons based on sample size calculation.

d.Why was a one-tailed test used?

Response: We thank the reviewer for bringing this to our attention. This was a typo and we have corrected it. In fact, two-tailed test was used by us.

 The one-tailed formula of 1-Way ANOVA is as follows, and it is different from the formula in the article:

In the above formula,κ=nA/nB is the matching ratio, σ is standard deviation, σA is standard deviation in Group "A", σB is standard deviation in Group "B", Φ is the standard Normal distribution function, α is Type I error, τ is the number of comparisons to be made, β is Type II error, meaning 1−β is power.

  1. Line 130: “Mote” should be “Motor.”

Response: Thank you for the suggestion. We have corrected it in the revised manuscript (Line 149).

  1. Line 152-163. More details of the MRI paradigm are needed. E.g., what was the field strength?

Response: Thank you for the suggestion. We have added relevant detail of the MRI in the revised manuscript (Line 175-178).

Our modifications are as follows:

To compare changes in functional connectivity mediated by rTMS, each participant was scanned on a 3.0-T MRI scanner (Discovery MR750, GE Medical Systems, Milwaukee, WI) with a standard 8-channel head coil. These participants were scanned T1-weighted and rs-fMRI acquisitions, within 24 hours before and after the intervention.

  1. Line 186-187. Why was no adjustment for multiple statistical tests performed?

Response: We thank the reviewer for raising this valuable concern. Multiple comparison was performed in this study. We have corrected it in the revised manuscript.

Our modifications are as follows:

Patients were compared between groups with one-way ANOVA with least significant difference post-hoc analysis correcting for multiple comparisons [25], (Line 212-213)

  1. Lines 192-193. What does “as appropriate” mean? Why was one not used exclusively?

Response: We thank the reviewer for raising this valuable concern. We realize that there will be misunderstandings in our statements. In fact, these two methods were used in statistics respectively according to the difference of the smallest item. In order to avoid misunderstandings, we have modified the expression in the Line 217-219. In fact, stroke site, Current smoking and Current alcohol drinking were used Fisher's exact test. While, Age, Sex, Affected hemisphere, History of diabetes, and History of hypertension were used chi-square test.

  1. Tables 2-3. This format is unclear, what is being shown here? Please elaborate more in the text.

Response: Thank you for the suggestion. We have corrected it in the revised manuscript (Line 241-267).

Reviewer 2 Report

Comments and Suggestions for Authors

The work is interesting, and the proof is understood, but there are some comments,

"It is advisable to follow the format, as there are sections, such as the abstract, where it is no longer necessary to write the words "method, result, etc".

in the question of the formulas, they are not numbered and the meaning of each of the variables is not specified. In order to understand the context and the results, it is also necessary to include the numbers of the equations and the meaning of each variable.

It is suggested that the section on motor function assessment should be explained in more detail. It only gives the references, but does not explain how they are used in your research.

on the other hand, the discussion is very broad, but the conclusions are very short, it is necessary to detail a little more about the procedure of analysis and treatment.

I think that the limitations that they write at the end of the discussion section compromise the research because they are conditions that could be mentioned from the beginning to understand that the results only apply and are limited to the case study they propose. The last statement of the discussion could refer to future work. The conclusions are not a reflection of all the results that are presented.

Comments on the Quality of English Language

The wording aids the better understanding of the document. The writing is correct in English.

Author Response

1, It is advisable to follow the format, as there are sections, such as the abstract, where it is no longer necessary to write the words "method, result, etc".

Response: Thank the referee for the valuable suggestion. We have made modifications in the abstract.

2, In the question of the formulas, they are not numbered and the meaning of each of the variables is not specified. In order to understand the context and the results, it is also necessary to include the numbers of the equations and the meaning of each variable.

Response: Thank you for the suggestion. According to your prompt, we have made modifications in the Lines 131-143.

Our modifications are as follows:

As the study involved four groups, the calculating formula of sample size was as follows:

In the above formula, n is sample size, σ is standard deviation, Φ is the standard Normal distribution function, α is Type I error, τ is the number of comparisons to be made and β is Type II error, meaning 1−β is power. Based our sample size calculation on the Fugl-Meyer Assessment for upper extremity (FMA-UE) results of previous pre-experiments that = 11 (average of group A), = 18 (average of group B), σ = 6.0,  = 3, α = 0.05, and β = 0.20.

3, It is suggested that the section on motor function assessment should be explained in more detail. It only gives the references, but does not explain how they are used in your research. On the other hand, the discussion is very broad, but the conclusions are very short, it is necessary to detail a little more about the procedure of analysis and treatment.
Response: We thank the reviewer for bring this to our attention. We have added more detail about motor function assessment in the Lines 151-154. At the same time, we add our conclusions in the Lines 403-413. 

4, I think that the limitations that they write at the end of the discussion section compromise the research because they are conditions that could be mentioned from the beginning to understand that the results only apply and are limited to the case study they propose. The last statement of the discussion could refer to future work. The conclusions are not a reflection of all the results that are presented.

Response:  Thank you for your concern. Indeed, our work may have certain limitations due to the design of the study. However, the major found of the study has confirmed that continuous rTMS treatment is not the best choice for patients over a period of six weeks. Through there was  clinical significance existing in our work, our further experiments including different stroke patients have been designed based on the current experimental results. We add our conclusions in the Lines 403-413.

Reviewer 3 Report

Comments and Suggestions for Authors

The authors of the paper, 'Determining the optimal stimulation sessions for TMS-induced recovery of upper extremity motor function post-stroke: a randomization controlled trial’ have tried to investigate optimal treatment sessions of TMS for upper extremity dysfunction after stroke during the 6-week treatment and examined the mechanism of the optimal treatment sessions by using motor evoked potentials and resting-state functional magnetic resonance imaging. The paper is well written except for one or two points that need to be rectified/corrected which are as

1.         Re-write the line 67-69 with more clarity.

2.         Write a line or two regarding how samples were randomized.

3.         In section 2.3, write a line regarding the source of the formula and                      define μA, μB, Z, Z1, Ï•, etc.,

4.          In Line 130, the first word should be 'Motor'

5.         Write a line regarding correcting significance for multi-comparisons.

Comments on the Quality of English Language

 Minor editing of English language required

Author Response

  1. Re-write the line 67-69 with more clarity.

Response: Thank the referee for the valuable suggestion. We have re- written it in the Lines 66-72.

Our modifications are as follows:

Though numerous rTMS-based clinical trials have been reported, there is still no consensus on its optimal therapeutic protocol. For example, rTMS is effective in improving upper extremity dysfunction, but what conditions can the maximum therapeutic effect be achieved still need to be further clarified in the treatment protocol including the number of rTMS pulses, treatment frequency, treatment sessions, etc. Among these,  treatments sessions is an vital influencing factor of which clinicians lack objective evidence to choose an suitable treatments sessions of rTMS.  ( Line 66-72)

  1. Write a line or two regarding how samples were randomized.

Response: Thank the referee for the valuable suggestion. We have made modifications in the Line 145-146.

  1. In section 2.3, write a line regarding the source of the formula and define μA, μB, Z, Z1, Ï•, etc.,

Response: Thank the referee for the valuable suggestion. We have made modifications on the formula of μA, μB, Z, Z1, Ï• in the Line 138- 143.

  1. In Line 130, the first word should be 'Motor'

Response: Thank the referee for the valuable suggestion. We have made modifications in the Line 145.

  1.  Write a line regarding correcting significance for multi-comparisons.

Response: We thank the reviewer for raising this valuable concern. Multiple comparison was performed in this study. We have corrected it in the revised manuscript.

Our modifications are as follows:

Patients were compared between groups with one-way ANOVA with least significant difference post-hoc analysis correcting for multiple comparisons [25], (Line 212-213)